# Disapproval from romantic partners, friends and parents: Source of criticism regulates prefrontal cortex activity

Michelle Jin-Yee Neoh[1], Atiqah Azhari[1], Claudio Mulatti[2], Marc H. Bornstein[3,4], Gianluca Esposito[1,2]*

1 Psychology Program, School of Social Sciences, Nanyang Technological University, Singapore, Singapore, 2 Department of Psychology and Cognitive Science, University of Trento, Rovereto, Italy, 3 Child and Family Research, Eunice Kennedy Shriver National Institute of Child Health and Human Development, Bethesda, MD, Untied States of America, 4 Institute for Fiscal Studies, London, United Kingdom

* gianluca.esposito@ntu.edu.sg

**Data Availability Statement:** The datasets generated during and/or analysed during the current study is available in the open access

## Abstract

The prevalence of criticism in everyday social situations, and its empirically demonstrated association with psychopathology, highlight the importance of understanding neural mechanisms underlying the perception and response of individuals to criticism. However, neuroimaging studies to date have been limited largely to maternal criticism. The present study aims to investigate neural responses to observing criticism occurring in the context of three different relationship types: romantic partners, friends, and parents–from a third-party perspective. 49 participants were recruited and asked to rate the perceived criticism for these relationships. Functional near-infrared spectroscopy was used to measure changes in oxygenated haemoglobin levels in the prefrontal cortex when participants read vignettes describing three different scenarios of criticism. Participants were randomly assigned to 3 groups where the given description of the relationship of the protagonist to the source of criticism for each vignette was randomised. A significant interaction between relationship type and perceived criticism ratings for mothers was found in the dorsolateral prefrontal cortex. Compared to low perceived criticism, high perceived criticism individuals showed increased activation reading vignettes describing criticism from romantic partners and parents but decreased activation for those from friends. Findings contribute to understanding neural responses to criticism as observed from a third-party perspective. Future studies can look into differentiating neural responses of personalised experiences of criticism and third-party observations.

## Introduction

Criticism, defined as negative evaluative feedback received from other people in social interactions [1,2] is often construed as unpleasant but is common, and often unavoidable, in social interactions of everyday life. It can thus be considered a naturally occurring and common form of social threat [3]. Negative emotional reactions towards criticism are considered

institutional data repository (DR-NTU) at the link: https://doi.org/10.21979/N9/JHHBXB.

**Funding:** GE received funding from Nanyang Technological University (NAP-SUG). https://research.ntu.edu.sg/featuredprofile/NAP/Pages/default.aspx The funders had no role in study design, data collection and analysis, decision to publish, or preparation of the manuscript.

**Competing interests:** The authors have declared that no competing interests exist.

normative [4] and frequently motivate behaviour adjustment [5]. Excessive criticism has been associated with negative individual outcomes and plays a role in the development and recurrence of psychopathologies, such as depression [6,7] and schizophrenia [8]. Notably, maternal criticism predicts depression onset in children [9]. The link between criticism and relapse of psychopathology is also supported by extensive empirical research conducted on the expressed emotion (EE) construct, which is a measure of the family environment, reflecting the degree of criticism, hostility, and emotional over-involvement characterising close family members of a psychiatric patient [10]. Criticism is the most important element in EE and has been consistently linked with poor clinical outcomes [11]. Taken together, the role and importance of criticism in shaping behaviour and empirically supported associations of criticism with individual social outcomes and mental well-being motivate deeper understanding of possible neural correlates of criticism.

## Neural correlates of criticism

From existing studies examining typically developing adolescents as they received maternal criticism, three related processes have been proposed to be involved in neural responses to maternal criticism [12]–(i) emotional reaction, (ii) regulation of generated emotion, and (iii) social cognitive processing (i.e., mentalising or perspective taking in understanding the mental state of the criticism source). Increased activity in affective networks (putamen, insula) and decreased activity in cognitive control networks (dorsolateral prefrontal cortex; dlPFC) and social cognitive networks (temporoparietal junction and posterior cingulate cortex) have been observed when hearing one's own mother's criticism [12], implying that neural responses to maternal criticism carry increased emotional reactivity but decreased cognitive control and social cognitive processing.

First, with regard to (i) emotion reaction, negative emotional reactions to criticism are considered normative as mentioned above [4] such as feelings of hurt. Negative emotional reactions have been associated with activity in both prefrontal regions such as the rostral anterior cingulate cortex (rACC) [13–15] and subcortical-limbic regions such as the amygdala [16,17]. There has also been some initial evidence pointing towards amygdala hyperactivation and prefrontal hypoactivation in individuals high on perceived criticism (PC) who listened to maternal criticism [18]. This pattern of activation is indicative of the implication of a neurocircuit related to dysfunctional emotional regulation and depression vulnerability [19–21]. Sustained brain activity in affective networks was also observed during the presentation of negative stimuli and subsequent rest periods [18,22,23]. In addition, the negative emotional reaction accompanying the experience of criticism is likely to influence processing of social and emotional information (i.e. the criticism) and decision making in social situations [24]. The intensity of the emotional reaction can influence attentional deployment and meaning attributed to the situation, thereby cueing different response types to the encountered situation.

Second, with regard to (ii) emotion regulation, the process model of emotion regulation proposes the conceptualisation of emotional responding at behavioural and physiological levels as being products of both emotional reactivity and emotion regulation [13]. Emotion regulation refers to processes by which individuals influence which, when, and how emotions are experienced and expressed. Such cognitive processes include reappraisal and suppression that downregulate negative affect [25]. Cognitive reappraisals regulate emotional responses [26] and have been shown to modulate self-reported emotional experience [27]. Emotion regulation has been associated with enhanced activity in frontal regions of the dlPFC, orbitofrontal cortex (OFC), ventrolateral prefrontal cortex (PFC), and anterior cingulate cortex (ACC) [28–31]. These frontal regions provide top-down inhibition of subcortical limbic circuits, such as the amygdala and hippocampus, responsible for emotion generation [30].

Third, in the context of criticism, (iii) social cognitive processing involves having a sense of where the source of criticism is coming from, which encompasses mentalising and perspective taking. A number of brain regions including the dorsomedial PFC, posterior superior temporal sulcus, and temporoparietal junction are involved in social cognitive processing [32–35], whereas the ventromedial PFC and posterior cingulate cortex have been implicated in thinking about close others' minds and self-related processing [36,37].

While most of the studies discussed above have been conducted on self-referential criticism, few studies have been conducted on the neural response during the observation of criticism occurring in the social interactions between others–other referential criticism. These few functional magnetic resonance imaging (fMRI) studies also focused only on examining the (i) medial PFC due to its engagement in self-processing and mentalising about others' states and (ii) amygdala due to its influence on attention to emotional-expression stimuli as regions of interest. One study found that individuals with generalised social phobia showed significantly increased blood oxygen level dependent (BOLD) responses in the dorsal medial PFC and the amygdala compared to the control group in response to self-referential criticism but not to other-referential criticism [38]. Similar fMRI findings in another study have also indicated activations in prefrontal regions such as the ventral and dorsal medial PFC, posterior cingulate cortex, inferior parietal lobule and temporal poles during self-referential processing compared to other-referential processing of criticism stimuli [39]. In addition, studies comparing self-referential and other-referential processing of valenced stimuli have also found evidence of differences in the activation patterns between self and other-referential processing. Activation in the ventral and dorsal anterior medial PFC for trait adjective judgments targeting the self while activation in the posterior dorsal medial PFC was observed when adopting a third person perspective in making trait adjective judgments about another person [40]. Different activation patterns were also observed reading valenced trait adjectives associated with either the self or other [41].

Given involvement of these parts of the PFC in emotion reactivity and regulation, coupled with previous findings of decreased prefrontal control to criticism, the present study sought to investigate the neural correlates of responses to other referential criticism specifically in the PFC using functional near-infrared spectroscopy (fNIRS).

## Criticism and perceived criticism

It has previously been suggested that affective social factors, such as the perceived criticism (PC) of the source, affects neural engagement in processing maternal criticism [12]. PC is a subjective measure of the level of criticism in an individual's closest or most meaningful relationships usually a romantic partner, spouse or parent. It was initially developed by [42] as a simplified measure for the expressed emotion construct and has been described as reflecting the amount of criticism that "gets through" to patients [42]–a high rating thus indicates a high amount of criticism "[getting] through" to the individual in the particular relationship being rated. This means that PC may be highly related to both the objective amount of criticism in the individual's social environment and the individual's experience of the relationship with the target of the PC rating. In this construal, PC is representative of both objective and subjective experiences of criticism [43]. PC ratings reflect perceptions of destructive criticism as opposed to constructive [44]. PC ratings for parents or romantic partners who lived with the participants predict changes in depressive symptoms, whereas those of friends, influential figures, and the most critical individual to each participant do not [45].

Neuroimaging research into PC has revealed differences in neural responses to criticism between people with different levels of PC. In a study by [18], individuals were classified as

high and low on PC based on a median split. Findings from this study found that high PC individuals showed greater and sustained activation in the amygdala as well as reduced and less prolonged activation in the dlPFC to maternal criticism [18]. This activation pattern in the dlPFC is indicative of increased emotional reactivity and decreased cognitive control in high PC individuals to maternal criticism compared to low PC individuals. In addition, research demonstrating impaired cognitive processing and control of negative emotional information in individuals with high PC [46] is consistent with findings in [18] of less dlPFC activation in high PC individuals compared to low PC individuals. Individuals who rated their key relationship with a relative or person who was currently the most emotionally important to them–someone they shared the closest relationship with–as high in PC had greater difficulty exerting attentional control over negative emotional information in an experimental cognitive task, suggesting that they encounter greater difficulty in shifting attention away from negative emotional information [46] also found that high PC individuals were more likely to report hearing negative words than neutral words when presented with ambiguous blends of similar words differing only in one phoneme. This finding indicates that these individuals show a negative interpretation bias–misinterpreting ambiguous emotional information.

As it appears that PC ratings are highly related to the social environment and the criticism present in relationships with close others and picking up on individual differences in how negative information is processed at a neural level, we can expect individuals high on PC to interpret negative social information differently. Through emotional socialisation, individuals learn to express, understand, and regulate emotion during childhood [47], and these abilities persist in shaping social interactions of children [48]. These early childhood experiences also influence how an individual perceives social information. According to social information processing theory, individuals make use of processed social information together with experienced previous interactions to make sense of and approach social situations [49,50]. Based on previously learned knowledge and experience, individuals develop functional schemes–beliefs and expectations about how interpersonal relationships work–which are used in perceiving and interacting with the world [51]. Moreover, social information processing can be influenced by an individual's emotional socialisation in a number of ways: (i) guiding attention to different aspects of the situation, (ii) altering interpretations of situational cues, and (iii) determining subsequent behaviours [52,53]. The influence of emotional socialisation on social information processing suggests that prevalence of criticism in an individual's relationship with his/her parents may influence how individuals view and process social situations involving criticism differently. Because parents and their interactions with children play a significant role in the socioemotional development of children, it is expected that PC ratings of parents would relate to how individuals perceive criticism occurring in social situations. On this basis, we expected that individuals high on PC would demonstrate different neural responses from those with low on PC.

## Criticism and interpersonal relationships

A majority of previous neuroimaging studies has focused on either maternal or self-criticism, where an experimental paradigm, known as the maternal feedback challenge, is commonly adopted [54]. The maternal feedback challenge involves participants listening to recordings of maternal verbal criticism or praise. However, currently, little is known about criticism experienced in other social relationships, such as between romantic partners and friends. Individuals are deeply linked to others in their social environment, of which the ties with romantic partners, friends, and parents are viewed as the most crucial [55], reflecting the importance of the need for research investigating criticism occurring in these relationships as well. It is possible

that criticism originating from different sources–romantic partners, friends, and parents–is interpreted differently by individuals depending on their view of these sources. First, individuals develop mental representations of their relationships with others, which are conceptualised as representations of a particular type of relationship and the self and the partner in that particular relationship. These views of the relationship are expected to guide an individual's behaviour and serve as the basis for predicting and interpreting the other's behaviour [56], suggesting that individuals may interpret criticism differently depending on the relationship they have with the source of the criticism. Second, individuals may perceive criticism even when a critical intention is absent or in situations where observers disagree that the statement is critical (i.e., overperceive criticism in close relationships), which has been labelled as a criticality bias [57]. There is a positive correlation between an individual's criticality bias and negative attributions made about the behaviour of a counterpart in a relationship [58]. Consequently, individuals who tend to make more negative attributions about a counterpart's behaviour or events in the counterpart's life may be more likely to interpret their behaviour as destructive criticism [44]. These interpretations of the causes of counterparts' behaviours subsequently play a role in determining whether a behaviour is perceived as critical [57].

As such, the attributions made regarding the nature of the criticism that can influence an individual's reaction to the criticism. How another person's behaviour is appraised is central to both an individual's emotions, such as feelings of hurt in the case of criticism, and the perceived impact of the behaviour on the relationship. For example, individuals who perceive a particular comment to be intentionally hurtful distanced themselves from the source of criticism and hurt as well as felt greater emotional pain than if the comment were to perceived to be unintentional [59]. In a similar vein, it has been argued that the nature of criticism as a face threatening act will also influence one's reaction. According to the face management theory proposed by [60], face is the desired social image created by an individual through interactions with others. A relationship-specific face [61] is enacted where an individual's face becomes inextricably bound in a shared relationship identity as relationships develop closeness. As a result, expressions of disapproval pose a threat to one's relationship specific face, potentially causing hurt feelings and perceived relational devaluation. A study found that perceptions of face threat were associated with emotional reactions such as feelings of anger, hurt, embarrassment and anxiety/depression as well as perceptions of fairness [62]. Additionally, findings from this study also suggested that the nature and rules governing different social relationships affected the relevance of negative face threats–threat to one's need for autonomy and desire to avoid impositions by others—for friends than for romantic partners. The study found that perceived threat to negative threat were associated with feelings of anger/hurt and damage to the relationship only in friends but not romantic partners [62]. Hence taking together differences in attributions, levels of perceived face threat and qualities of interpersonal relationships, we expect that individuals differing in PC may also have different responses to other referential criticism originating from different sources.

## Present study

As mentioned, majority of neuroimaging studies discussed above were investigating self-referential criticism with few studies looking at the neural correlates of other referential criticism. In addition, the studies on other referential criticism tended to look primarily at the mPFC as a region of interest and varied in terms of the identity of the "other" as the source of the other referential criticism. In the present study, we aim to investigate PFC activation during exposure to other referential criticism involving an "other" from three different relationship types–romantic partners, friends and parents.

Hence, the present study aims to contribute to existing literature by providing both novel fNIRS data of PFC activation during exposure to other referential criticism and a functional characterization of the mental processes underlying the neural response to criticism observed in the social interactions of others from a third-party perspective while comparing romantic partners, friends, and parents as sources of criticism. We investigated PC as a moderating factor in neural responses to observed criticism using fNIRS and we expect differences in the neural response to emerge when individuals are exposed to scenarios involving criticism between individuals. Based on previous findings indicating individual differences in processing negative information between individuals with different levels of PC, we hypothesised that individuals with high perceived criticism ratings will show decreased PFC activation compared to individuals with low perceived criticism ratings.

## Method

### Recruitment

Participants ($n$ = 50, mean age = 21.67, females = 25) were undergraduates recruited through word of mouth and advertisements and compensated with course credits or remuneration. Inclusion criteria were 18–25 years of age without any reading disabilities or difficulties with English, as the experimental stimuli required reading. Having a psychiatric disorder was an exclusion criteria for participants. The study was approved by the Psychology Programme Ethics Committee at the Nanyang Technological University and the research conducted in this study was performed in accordance with the guidelines set forth by this ethics committee.

### Experimental procedure

Informed consent was obtained from all participants before the experiment was conducted. Participants were required to complete a pre-experimental questionnaire. Participants then proceeded to the experimental room where they were fitted with a NIRS cap, and the NIRS signal was calibrated.

During the experiment, participants were presented with vignettes describing scenarios involving criticism (Fig 1). NIRS recordings were made throughout the experiment, and NIRStim software was used to present these vignettes on a laptop. Participants were informed that they would be reading vignettes involving interactions between people and a description of the relationship between the persons involved will be given. Three different vignettes were used in the study where each vignette described a different scenario of criticism in a social

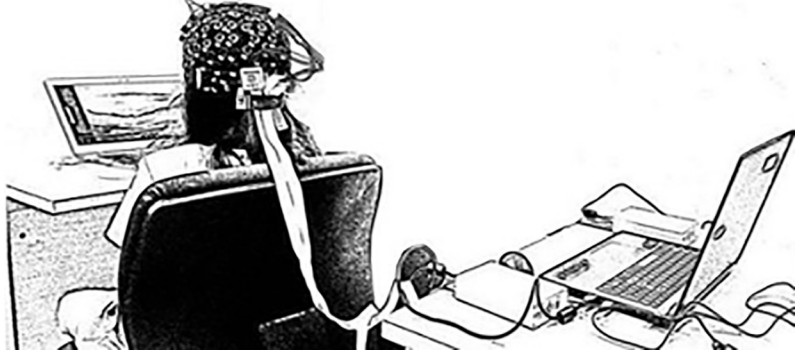

**Fig 1. Digital rendering of experimental setup depicting (i) 20 channel setup, (ii) NIRS device and (iii) laptop placement.**

situation. The three vignettes were shown in the same order to all participants. Before each vignette was shown, a brief description of the relationship between the protagonist and the source of criticism was shown to the participants. The order in which the relationship described was presented for each scenario of criticism was counterbalanced as: (a) Romantic Partner-Friends-Parents, (b) Friends-Parents-Romantic Partner, and (c) Parents-Romantic Partner-Friends. Participants were then randomly assigned to groups where they viewed the vignettes with the accompanying relationships' descriptions in one of these orders. Each vignette was presented for 90 sec, and the offset of each vignette was followed by a fixation point displayed in the centre of the blank screen (Fig 2).

After each fixation point before the onset of the next vignette, participants were asked to rate (i) the level of justification and (ii) the impact of the criticism for each vignette on a 10-point scale. This procedure served as a manipulation check to ensure that criticisms in the

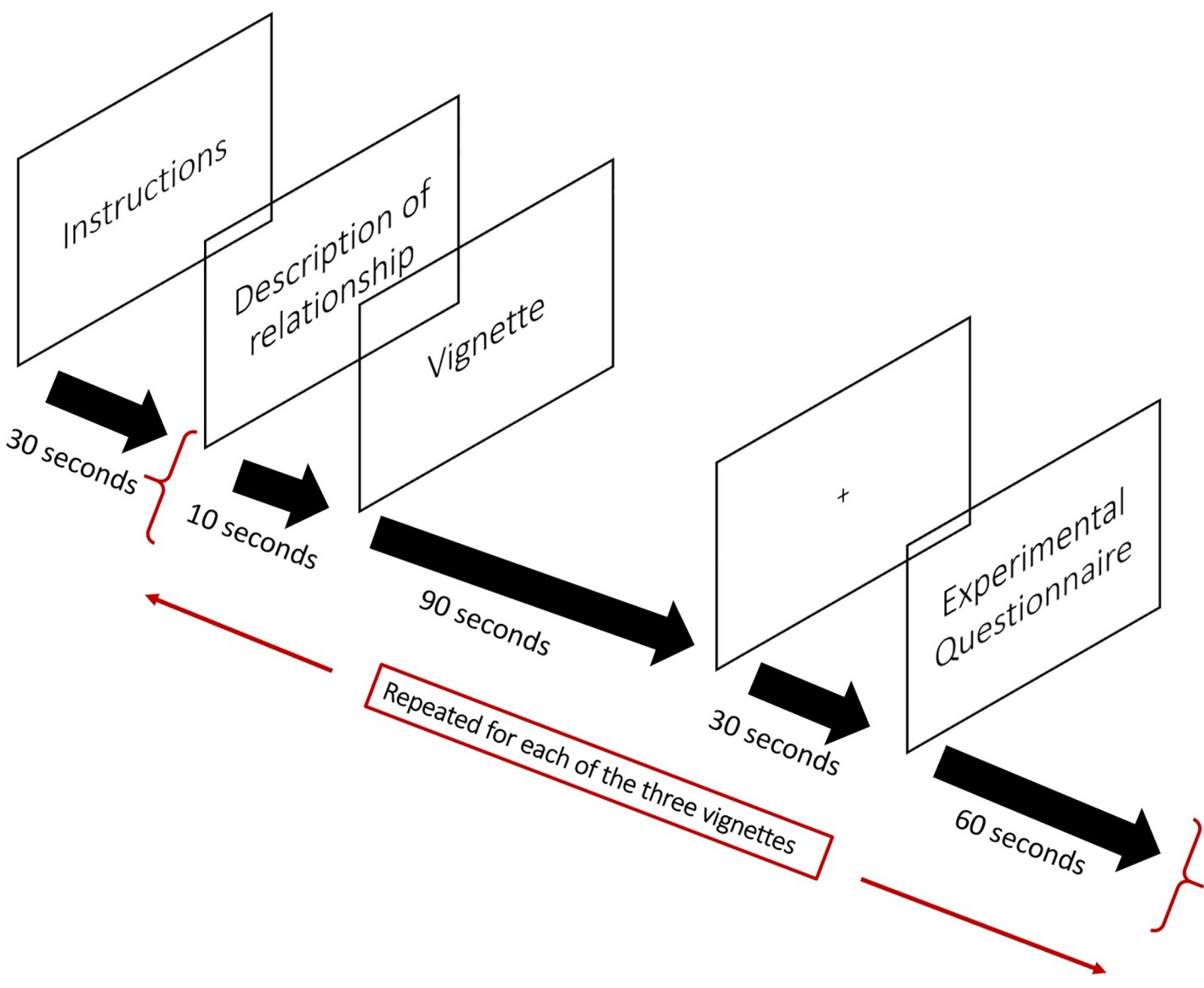

**Fig 2. Diagram of experimental stimulus depicting what was displayed.**

## Questionnaires

The pre-experimental questionnaire consisted of questions about demographics and Perceived Criticism (PC) ratings. Demographic information–age, gender, nationality–of participants was collected along with information about social relationships: number and age of siblings and romantic relationship status and history. PC ratings for each of target relationship: (i) romantic partner, (ii) friend and (iii) parents; mother and father of the participant were evaluated with the question "How critical is (the relative) of you?" which was rated on a 10-point scale [42] (Table 1). Table 1 indicates the correlations between the PC ratings of the three relationship types. PC ratings have high predictive validity, correlate with expressed emotion [42,63], and enjoy high test-retest reliability [42].

## Experimental stimuli

Three vignettes depicting different scenarios of constructive criticism involving the protagonist who is with either his/her (i) romantic partner, (ii) friend or (iii) parents were constructed. Each vignette was accompanied by a brief description of the relationship between the protagonist and the source of criticism. Measures were taken to ensure the authenticity and identification with the protagonist in the situations presented in the vignettes. First, the vignettes were adapted from Shame Situation vignettes of the Situated Emotion Experience Questionnaire (SEEQ) [64]. The SEEQ maximises the ecological validity and cross-cultural representativeness of shame experiences through an extensive bottom-up sampling method involving United States and Japanese samples. Thus, the material stayed as close as possible to the everyday ecology of experienced shame. In addition, male and female versions of the vignettes were used with boys and girls, respectively, and differed only in the name used for the main protagonist to facilitate participants' identification with the protagonist. Second, a common structure was adopted for each vignette. Each vignette consists of a brief context outlining a particular event warranting criticism of the protagonist's actions, as adapted from the situations described in the Shame Situation vignettes which were appraised to be the most relevant to the experience of shame [64]. Constructive criticism was then presented in the form of an explicitly stated character flaw and a call for change in the protagonist in a block quote to maximise salience. All vignettes were approximately 120 words and were presented for 90 seconds. An example of a vignette for a male participant is as follows:

> *Chris performed above expectations in exams in the first semester of the school year. As a result, he got complacent and neglected his studies in the next semester. He was lazy and did not put effort into preparing and studying for the next semester's exam despite his parents expressing their concern. Consequently, he did very poorly for the second semester and upon learning of his results, Chris' parents admonished him:*

**Table 1. Table of correlations between PC ratings for each target relationship.**

| PC ratings | Romantic partner | Friend | Mother |
|---|---|---|---|
| Friend | 0.82 ($p < .001$) | 1 | |
| Mother | 0.71 ($p < .001$) | 0.63 ($p < .001$) | 1 |
| Father | 0.51 ($p < .05$) | 0.46 ($p < .01$) | 0.58 ($p < .001$) |

*"Chris, this incident shows your complacency and laziness. You should have put in consistent effort and commitment towards self-improvement. This is something that we think you should change. We are disappointed in your attitude this time. Hopefully, this incident will serve as an opportunity for you to learn and change."*

## fNIRS recording

fNIRS recordings were made with the functional NIRS imaging system (NIRSport, NIRx Medical Technologies LLC, Glen Head, NY, USA) which operates using light of wavelength 760-850nm. This system measures relative oxygenated hemoglobin (HbO) and deoxygenated hemoglobin (Hb) concentrations which indicate cerebral activation and deactivation. fNIRS allows monitoring of local blood oxygenation where more active brain regions exhibit greater concentrations of HbO. The NIRS device consists of LED-sources (emitting optode) that transmit long-wave light to cortical tissues and detectors (receiving optode) that measuring the intensity of returning light. The optical signal was recorded at a sample rate of 7.81 Hz. In this study, the hemodynamic changes of the PFC were measured. The configuration of 8 sources and 7 detectors on the NIRS cap formed a 20 multi-distant channel setup where data from cortical measurements were recorded using the NIRStar Software 14.0 (see Fig 3 for channel locations and the corresponding positions in the PFC). The distance between sources and detectors did not exceed the optimal interoptode distance of 3 cm. Probes on the NIRS cap were adjusted at the start of the experiment before calibration of signal quality for each participant.

## NIRS data pre-processing

Preliminary signal processing was conducted using the nirsLAB software (NIRx Medical Technologies LLC, Glen Head, NY, USA). Data pre-processing included (i) truncation of signals

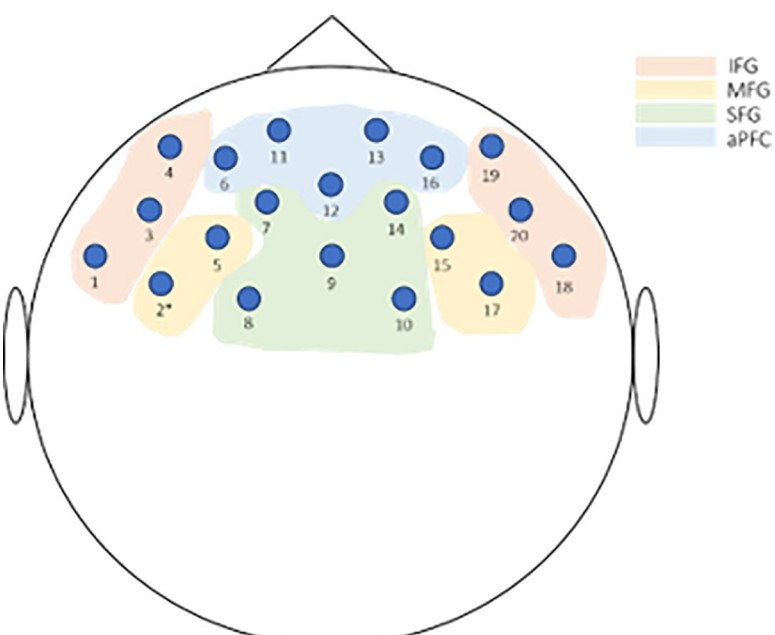

**Fig 3. Diagram displaying channel locations and corresponding positions in the inferior frontal gyrus (IFG), middle frontal gyrus (MFG), superior frontal gyrus (SFG), and anterior PFC (aPFC).** Montage for probe placement is based on the 10–20 system.

recorded outside of the stimulus timeframes, (ii) removal and replacement of spike artefacts with nearest or random signals, (iii) removal of discontinuities, (iv) interpolation of channels, and (v) application of a band-pass frequency filter (0.1–0.2 Hz) to remove both very rapid (drifts) and slow (noise) fluctuations before hemodynamic states were computed. Pre-processed signals were then converted into the relative change in concentrations of HbO and Hb for each channel using the modified Beer-Lambert law–the assumption of a linear relation between the absorption of light and concentration of brain tissue (see Fig 4).

For each participant, visual inspection of signals was conducted for each channel, and spike artefacts were removed. The NIRS data were analysed at two levels: within-subject and group-level. First, a general linear model (GLM) was run for each participant where individual beta coefficients were calculated for each level of relationship type: romantic partners, friends, and parents. Second, group level analysis involved aggregating these beta-coefficients from the HbO GLM of each participant into a group-level GLM.

All results were subjected to Bonferroni correction (p < .05) and were depicted in a topographical map with probe labels allowing for the subsequent mapping of brain regions. Given that the signals recorded are unlikely to be independent of each other as they represent the response of a single brain, false discovery rate (FDR) correction was also applied to account for multiple comparisons across the 20 channels.

## Data availability

The datasets generated during and/or analysed during the current study is available in the open access institutional data repository (DR-NTU) at the link: https://doi.org/10.21979/N9/JHHBXB.

## Analytic plan

First, the descriptive statistics of the participants' ratings of the level of justification and impact of the criticism vignettes were calculated. This is to determine if the criticism vignettes were viewed by the participants as relatively authentic situations involving criticism. A one-way, repeated measures analysis of variance (RM-ANOVA) for relationship type was conducted to

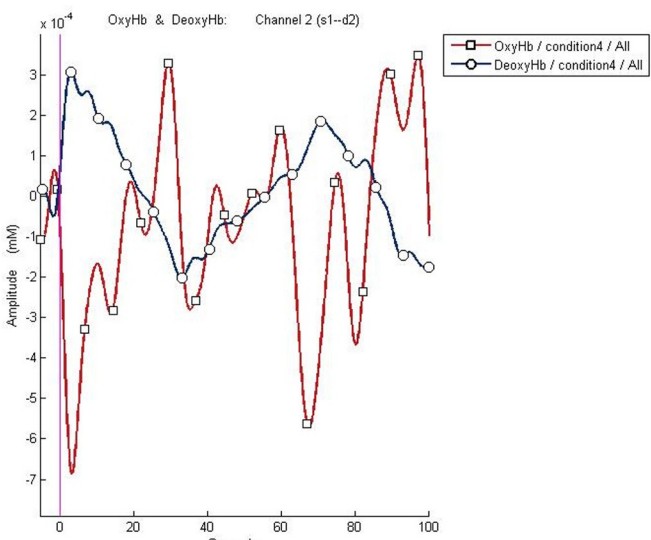
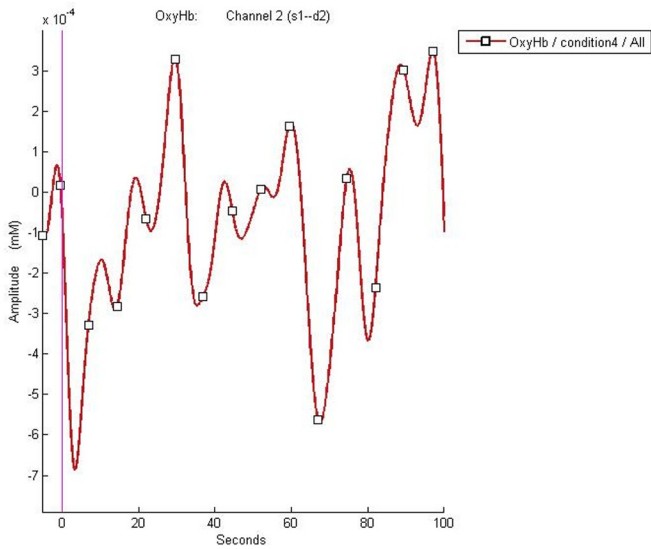

**Fig 4.** Processed HbO/Hb concentration (left) and HbO concentration (right) for one condition in a single NIRS channel.

**Table 2. Table of means for ratings of level of justification and impact of criticism.**

| Relationship type | Level of justification | | Impact | |
|---|---|---|---|---|
| | Mean | Standard deviation | Mean | Standard deviation |
| Romantic partner | 7.51 | 2.05 | 7.04 | 2.12 |
| Friends | 7.12 | 2.12 | 6.33 | 1.95 |
| Parents | 7.55 | 1.82 | 7.06 | 1.93 |

investigate whether there were differences in ratings of the level of justification and impact of the criticism vignettes based on the relationship type of the source of criticism.

For the fNIRS data, a preliminary data analysis was conducted to check for any gender differences in the neural response to the criticism vignettes. A two-way, mixed ANOVA will then be conducted on the processed changes of HbO concentrations in order to test our hypothesis. The results of the ANOVA will indicate if there are significant differences in the neural response to criticism between individuals differing in PC.

## Results

Data collected in the pre-experimental and experimental questionnaires were analysed using SPSS Version 23.0. The final sample used for analysis consisted of 49 participants with M age = 21.65, SD = 1.49 (female = 25), of which 27 currently or previously had been in a romantic relationship. Data for one additional male participant were excluded at the pre-processing stage due to poor signal quality.

### Experimental questionnaire

Descriptive statistics of the level of justification and impact which were rated on a 10-point scale indicate that the criticism in each vignette was viewed as relatively authentic (Table 2). Correlations between PC and the level of justification and impact are summarised in Table 3. All of the correlations were nonsignificant.

One-way, repeated-measures analysis of variance (RM-ANOVA) for relationship type was conducted on the ratings for justification and impact of the criticism in each vignette obtained in the experimental questionnaire.

**Perception of level of justification of criticism.** There was no significant difference among relationship types of the source of criticism for the ratings for how justified the criticism was ($F_{(2, 96)} = 0.80$, $p = 0.46$, $\eta_p^2 = 0.02$).

**Perception of impact of criticism.** There was no significant difference among relationship types of the source of criticism for the ratings for the impact of the criticism ($F_{(2, 98)} = 2.92$, $p = 0.06$, $\eta_p^2 = 0.06$).

### fNIRS results

From the preliminary analyses, there was no significant (i) main effect of gender or (ii) interaction of relationship type*gender on hemodynamic changes in the PFC found–i.e. no

**Table 3. Table of correlations between PC ratings and ratings of level of justification and impact of criticism.**

| PC ratings | Level of justification | Impact |
|---|---|---|
| Romantic partner | -0.003 | 0.18 |
| Friends | 0.07 | -0.10 |
| Mother | 0.17 | 0.22 |
| Father | 0.1 | 0.09 |

significant difference in HbO changes in any of the 20 channels between the relationship type conditions (romantic partners, friends, and parents). Hence, male and female samples were combined and analysed together.

Relationship type of the source of criticism (romantic partner, friend, parents) was the within-subjects factor and PC ratings for each target relationship (romantic partner, friend, mother, father) for each participant from the pre-experimental questionnaire were considered as between-subjects factors in a two-way, mixed ANOVA of relationship type * PC ratings conducted on the processed changes of oxygenated haemoglobin levels (HbO) data to address our hypothesis.

## PC ratings

There was no significant main effect of PC ratings for any target relationship (romantic partner, friend, mother, father) on hemodynamic changes in the PFC found. A significant interaction between relationship type and the PC rating of mothers on hemodynamic changes in the PFC was found in part of the dlPFC in the left middle frontal gyrus (BA46L, Channel 2 (1 2); see Fig 3) ($F_{(2, 128)}$ = 7.69, $p$ < .05, Bonferroni corrected, $\eta_p^2$ = 0.03). This result remained significant after applying FDR correction.

In examining the interaction between relationship type and PC rating of mothers, the Pearson product-moment correlations were calculated between beta coefficients for each relationship type and PC rating of mothers (Table 4). The correlation coefficients are as follows: (i) Romantic Partner-PC (Mother) (RPM) was (r = 0.20, n = 44), (ii) Friend-PC (Mother) (FPM) was (r = -0.44, n = 43), and (iii) Parents-PC (Mother) (PPM) was (r = 0.26, n = 41). Fisher r-to-z transformation was applied to the correlation coefficients in order to examine the significance of the difference between the correlation coefficients by comparing the z scores (Table 4). The results summarised in Table 4 showed significantly different correlations for (i) the Romantic Partner-PC (Mother) and Friend-PC (Mother) correlations (Z = 3.02, p < 0.05) and (ii) Friend-PC (Mother) and Parents-PC (Z = -3.25, p < 0.05). The Romantic Partner-PC and Parents-PC correlations were not significantly different (Z = -0.29, p = 0.77 > 0.05). From Fig 5, it can be observed that as PC ratings for the individual's mother increased, activation of the left middle frontal gyrus of the dlPFC (BA46L) increased when reading the vignettes describing criticism from romantic partners and parents but decreased when reading the vignettes describing criticism from friends. This significant result suggests that PC ratings moderate neural response to criticism originating from different sources of different relationship types, resulting in different activation patterns observed between high and low PC individuals for criticism from the different relationship types.

**Relationship type.** A significant main effect of relationship type was found for (i) PC rating of close friends in part of the dlPFC in the left middle frontal gyrus (BA46L, Channel 1 (1 1)) ($F_{(2, 132)}$ = 5.78, $p$ = 0.012, Bonferroni corrected, $\eta_p^2$ = 0.08), (ii) PC rating of mothers in part of the dlPFC in the left middle frontal gyrus (BA46L, Channel 1 (1 1)) ($F_{(2, 132)}$ = 5.77, $p$ = 0.01, Bonferroni corrected, $\eta_p^2$ = 0.08), and (iii) PC rating of fathers in part of the dlPFC in the left middle frontal gyrus (BA46L, Channel 1 (1 1)) ($F_{(2, 132)}$ = 5.73, $p$ = 0.01 < .05,

**Table 4. Table of Fisher r-to-Z transformation test for the relationship type * perceived criticism (Mother) interaction effect.**

| Channel | Romantic Partner-PC (Mother) (RPM) | | Friend-PC (Mother) (FPM) | | Parents-PC (Mother) (PPM) | | RPM-FPM | | RPM-PPM | | FPM-PPM | |
|---|---|---|---|---|---|---|---|---|---|---|---|---|
| | *n* | *r* | *n* | *r* | *n* | *r* | *Z* | *p* | *Z* | *p* | *Z* | *p* |
| Channel 2 | 44 | 0.20 | 43 | -0.44 | 41 | 0.26 | 3.02 | 0.0025** | -0.29 | 0.77 | -3.25 | 0.0012** |

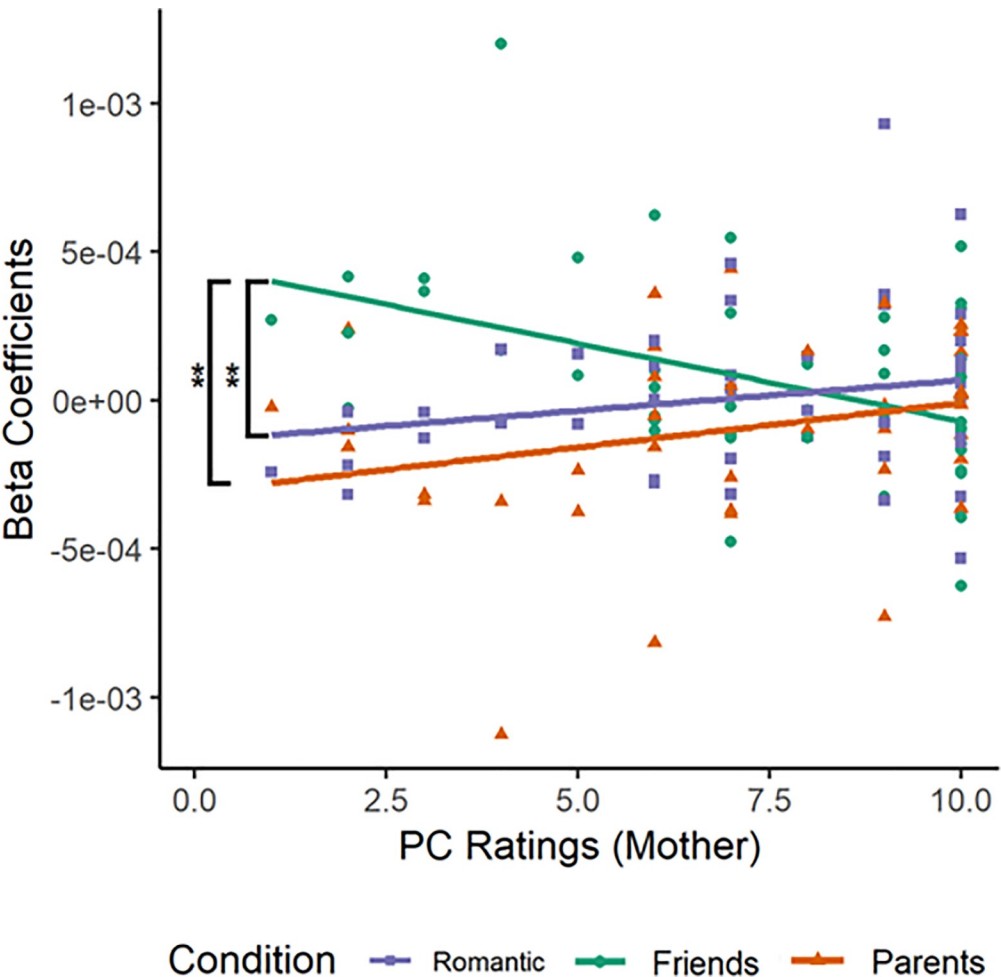

**Fig 5. Correlations between relationship type and PC ratings (Mother).**

Bonferroni corrected, $\eta_p^2 = 0.08$). However, after application of FDR correction to account for the multiple comparisons, these results were no longer significant. These results indicate that there were no significant differences in neural response to criticism originating from sources of different relationship types.

## Discussion

The present study attempted to examine differences in neural responses in the PFC to observed criticism in the social interactions of others between individuals with different levels of PC. A significant interaction between relationship type and PC ratings for mothers emerged. As PC ratings for the individual's mother increased, activation of the left middle frontal gyrus of the dlPFC (BA46L) increased when reading the vignettes describing criticism from romantic partners and parents but decreased when reading critical vignettes describing criticism from friends. However, the interaction found differs from our hypothesis in that the neural response in high PC individuals is characterised by increased cognitive processing as observed from the increased activation in the dlPFC, which is a part of cognitive control networks.

One possible explanation for this finding might be that there are differences in the way that criticism is processed by high PC individuals when the criticism is directly personally relevant as compared to when it is an indirect observation of criticism from a third-party perspective, as suggested by the findings from previous studies on self and other-referential processing of criticism and valenced stimuli. In relation to the neural processes involved in criticism mentioned earlier, the extent of the emotional response, emotion regulation and social cognitive processing may differ between processing criticism in the first-person compared to the third-person perspective. For example, a third person perspective could involve more social cognitive processes such as mentalising and empathy while also involving less intense emotional responses and regulation as mentioned earlier. Previous studies have indicated that high PC individuals decreased activation in response to maternal criticism [18]. Such decreased activation in cognitive control networks to maternal criticism might be associated with detachment from criticism to minimise cognitive processing related to the criticism due to the knowledge that this criticism may be hurtful [12]. Hence, it is possible that these results indicate a practised coping mechanism to detach from personally relevant criticism instead of cognitively processing the criticism, leading to a decreased activation of the dlPFC. Because the criticism described in the vignettes derived from a third-party perspective, high PC individuals may not engage in this coping mechanism and consequently, direct more attention and/or think more about the criticism described in the vignette, which may result in increased activation observed in the dlPFC. Another possible explanation for the significant interaction effect could be that the increased activation in high PC individuals compared to low PC individuals might indicate that high PC individuals are less able to detach from criticism. When criticism is not directly personally relevant, low PC individuals may minimise cognitive control processing of the criticism, whereas high PC individuals may be less able to do so possibly due to the negative interpretation bias [46] as well as a greater objective and subjective experience of criticism in their everyday interpersonal environment. Data in this study are not well positioned to support this interpretation; hence additional experimental research comparing conditions of personally relevant criticism and criticism from a third-party perspective is needed.

The interaction between relationship type and PC ratings for the mother on the neural responses to criticism is consistent with findings in [45] that only PC ratings for romantic partners and parents who lived with the participants predicted depressive symptoms. Hence, past findings that PC ratings were meaningful in reflecting the home environment could also be a possible explanation for the greater activation observed in the dlPFC for high PC individuals in reading the vignettes describing romantic partners and parents but not in friends. This observation could be a possible indication that high PC individuals are less able to detach from criticism originating from those target relationships that the PC ratings were most meaningful in reflecting. In other words, findings in the present study accord with the conclusion in [45] that PC ratings were most meaningful in reflecting the interpersonal home environment rather than that with friends or other people not living with the participant.

The significant interaction effect between relationship type and PC ratings for mothers found in the present study highlights the relationship of PC ratings with the perception of criticism and the corresponding neural correlates of criticism observed in the social interactions of others. This finding suggests that the social environment and experiences of an individual may influence how they view and process information regarding social situations and interpersonal relationships in others. It also raises the importance of investigating differences between the direct experience and the indirect observation of criticism especially in light of these initial findings on differences between high and low PC individuals. It also highlights the importance of individual factors–such as perceived criticism, personality, self-esteem, gender–in affecting the neural response to criticism, of which personality, self-esteem and gender have yet to be

examined in depth in relation to the neural correlates of criticism. Given that criticism is encountered on a regular basis in many, if not all, of an individual's interpersonal relationships and social environment, further research can be instrumental in better understanding the neural and cognitive processes involved in the perception and response to criticism.

In addition, findings from this study contributes to a greater understanding of the neural mechanisms involved in observing criticism from a third-person perspective and other-referential processing of criticism. Firstly, the significant interaction effect between relationship type and PC ratings indicate that the identity of the "other" implicated in other-referential processing can influence the neural response involved. In this respect, studies investigating self/other referential processing tend to either use generic third person pronouns (such as He/She; see [65] for the Visual-Verbal Self-Other Referential Processing Task) or only one specific type of "other" such as a close friend in [40]. Secondly, more areas of the PFC were studied in the present study, which suggest the possibility of other processes and prefrontal areas implicated in other-referential processing that have not been previously studied. Finally, understanding the processes involved when an individual observes criticism being directed at others can provide a clearer picture to how individuals navigate and respond during social interactions with others that involve occurrences of criticism directed at others (e.g. watching a friend's parent criticise him/her).

However, there were no significant main effects of the participant's PC ratings to the three different target relationships (romantic partners, friends, mothers and fathers (parents)). A possible explanation for these nonsignificant findings might reflect sample size. Only 27 participants in the sample were currently in or had been in a romantic relationship; hence, only a small proportion of PC ratings for romantic partners were available. While initial results indicated main effects of PC ratings of romantic partners and fathers, these results did not remain significant after FDR correction. Future research should condition analyses on adequate power.

However, there are several limitations in the present study. Firstly, only external sources of criticism were examined in the present study although another equally significant source of criticism is the self, where self-criticism can be thought of as relating to a form of negative self-judgment and self-evaluation [66,67]. Similarly, self-criticism is a clinically relevant construct where excessive self-criticism has been shown to associated with a range of psychological disorders such as mood disorder [66] and social anxiety [68]. It has also been proposed that self-criticism may stimulate the same neurophysiological systems as external criticism [66,69]. Hence, future studies can look into comparing the differences in neural responses between self and external criticism.

Secondly, there are limitations to the use of fNIRS. Although fNIRS has a higher temporal resolution than fMRI due to its rapid acquisition rate, it has a limited spatial resolution which is confined to cerebral cortex [70]. Hence, this study is unable to investigate the roles and activation patterns of other important subcortical regions such as the amygdala in the neural mechanisms of criticism, which has also previously been shown to be involved. In addition, only the prefrontal cortex was measured in this study. Future fNIRS studies may consider measuring more, if not all regions of the cortex. The experimental paradigm used in this study can also be replicated using fMRI in order to probe deeper cortical activation with higher spatial resolution.

Lastly, the sample for this study consisted of youths aged 18–25 and was largely made up of Singaporean undergraduates. However, there are different sociocultural norms and expectations for the kinds of behaviour that warrant criticism as well as the general levels of criticism [71]. Hence, this study can be replicated across different cultures considering these cultural differences. For example, studies have shown that Japanese people tend to exhibit self-critical

tendencies in comparison to Europeans and/or Americans such as accepting negative self-relevant information more readily [72,73]. In addition, US Americans responded more assertively to criticism compared to Asian counterparts (Japanese, Chinese) [74]. Although the present study was conducted only on healthy individuals, future studies can look to investigate whether there are differences between individuals diagnosed with psychiatric disorders, such as mood disorders and social anxiety, and healthy individuals in terms of processing criticism observed in the social interactions of others.

## Conclusions

In conclusion, findings from the present study suggest that criticism perceived in an individual's interpersonal environment can influence how one processes criticism observed from a third-party perspective. This provides preliminary evidence that points toward differences in social information processing as a function of one's own interpersonal environment and past experiences, specifically in the context of observing criticism as it occurs in social interactions of others. Given the multitude of factors that shape an individual's interpersonal environment and the experiences one derives from this environment, it follows that there remains much to be elucidated about how such factors and individual differences would influence social information processing and consequently, how one views and processes observations of criticism as they occur in the interactions of others.

## Acknowledgments

We would like to acknowledge members of the Social Affective Neuroscience Lab at NTU for their assistance in the completion of this project.

## Author Contributions

**Conceptualization:** Michelle Jin-Yee Neoh, Atiqah Azhari, Gianluca Esposito.

**Data curation:** Michelle Jin-Yee Neoh, Atiqah Azhari.

**Formal analysis:** Michelle Jin-Yee Neoh, Atiqah Azhari.

**Investigation:** Michelle Jin-Yee Neoh.

**Methodology:** Michelle Jin-Yee Neoh, Atiqah Azhari.

**Writing – original draft:** Michelle Jin-Yee Neoh, Atiqah Azhari.

**Writing – review & editing:** Michelle Jin-Yee Neoh, Atiqah Azhari, Claudio Mulatti, Marc H. Bornstein, Gianluca Esposito.

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
