## [Decision Letter · Decision Letter 0]

26 May 2020

PONE-D-20-03038

Disapproval from romantic partners, friends and parents: source of criticism regulates prefrontal cortex activity

PLOS ONE

Dear Dr. Esposito,

Thank you for submitting your manuscript to PLOS ONE. After careful consideration, we feel that it has merit but does not fully meet PLOS ONE’s publication criteria as it currently stands. Therefore, we invite you to submit a revised version of the manuscript that addresses the points raised during the review process.

We look forward to receiving your revised manuscript.

Kind regards,

Wi Hoon Jung, PhD

Academic Editor

PLOS ONE

Journal Requirements:

Reviewers' comments:

Reviewer's Responses to Questions

**Comments to the Author**

1. Is the manuscript technically sound, and do the data support the conclusions?

Reviewer #1: Partly

Reviewer #2: Partly

2. Has the statistical analysis been performed appropriately and rigorously? 

Reviewer #1: Yes

Reviewer #2: N/A

3. Have the authors made all data underlying the findings in their manuscript fully available?

Reviewer #1: Yes

Reviewer #2: Yes

4. Is the manuscript presented in an intelligible fashion and written in standard English?

Reviewer #1: No

Reviewer #2: Yes

5. Review Comments to the Author

Reviewer #1: This manuscript describes an interesting study towards the neural correlates of different sources of observed criticism. Participants (N=49, undergraduate students) were asked to indicate their personal perception of criticism (PC), which was then associated with neural responses to several vignettes on observed criticism, as measured by fNRIS. Findings show a moderating effect of PC on neural responses: those who rated PC higher showed more dlPFC activation when reading vignettes about romantic partners and parents, but not friends.

Although the topic of the study is certainly interesting and a possible strong addition to the existing literature, the authors seem to switch between self-experienced criticism and observed criticism throughout the manuscript. This makes it difficult to understand what it is exactly that the authors have investigated, and how their findings contribute to existing literature. A reframing of the topic would certainly improve the manuscript. Below I have outlined several additional concerns and questions that I would like to see addressed in a revision of this manuscript:

Abstract

1. The sentence “Perceived criticism ratings for these relationships from 49 participants were collected” if formulated a bit odd and could be rephrased.

Introduction

2. The authors do a good job explaining the neural correlates of emotion regulation and social cognitive processing, but information on the neural correlates of emotion reaction is scarce. Additionally, what exactly is the role of emotion reaction in the described process from experiencing criticism to emotional responding?

3. The introduction mainly contains literature on how the experience of criticism can shape later relationships and how criticism can affect neural processing. Given that the participants did not experience criticism themselves but merely observed criticism on someone else, literature should be added to describe results from earlier studies on observed or third-party criticism.

Methods

4. Were all participants recruited in the same way? As the authors describe “undergraduates compensated with course credits and undergraduates recruited through word of mouth and advertisements”

5. Were participants also screened on underlying psychiatric disorders, given the strong link between perception of criticism and psychopathology?

6. First the authors state that vignettes were shown in the same order to all participants, but then it is explained that participants are randomly assigned to one of three groups with different orders in the vignettes. Does this mean that the vignettes were not shown in the same order to all participants?

7. In the section “Questionnaires” the authors wrote “romantic partner, (ii) friend, parents; mother and father”. I assume parents should be a separate, third category.

8. Table 1: correlations appear to be significant, but statistical information (p-value) is missing.

9. Table 1: Are the correlation coefficients for mother and father significantly different from each other? If this is indeed the case: should this be taken into account, given that in the vignettes parents are collapsed (mother & father)?

10. Did every participant only see 3 vignettes (1 about romantic partner, 1 about friend, 1 about parents)? If so, how reliable are the fNIRS results if they are only based on one trial in each condition?

11. What exact instructions did the participants receive? Were participants instructed to identify themselves with the protagonist of the vignette?

12. Given that amount of perceived criticism was also a measure of interest, did vignettes differ in level of criticism?

13. Were PC ratings (resulting from the questionnaires) also correlated with level of justification and impact of the criticism for each vignette?

14. How was the 10-point scale for justification and impact formulated? (1 = not justified, 10= very justified)? And how can these scores indicate the authenticity of the vignettes?

Results

15. What exactly was investigated in the Pearson product-moment correlations? PC from Mother with what?

Discussion

16. Throughout the manuscript the authors seem to switch between self-experienced criticism and observed criticism. It should be made clearer what processes were actually studied and how the findings from the current study can be related to earlier work on self-experienced criticism.

17. A final concluding paragraph is missing.

Reviewer #2: Reviewer Form - The reviewer form is broken into the following parts:

In their article Disapproval from romantic partners, friends and parents: source of criticism regulates prefrontal cortex activity the authors report results from a study in which they used fNIRS to measure the brain response to criticizing messages (vignettes) coming from different (hypothetical) sources.

The topic of the paper is interesting and I believe this type of research is timely, the approach innovative, and the research has generally been conducted well. However, I had some difficulty to understand the design and results, and that made it really difficult for me to evaluate the manuscript. The authors might be able to address this in a revision, although I am not completely sure (since I didn’t fully understand the results).

1) Theoretical embedding/strategy: As said, I am generally favorable towards the approach. The use of fNIRS is novel and has potential. Also, I am aware that working in a novel area often goes along with a lack of prior theoretical work, so necessarily you have to go a bit of a data-driven route. That is fine with me, although I do think it would be worthwhile to dig a little deeper into the work that exists on these topics. E.g. in the communication science literature there has been a lot of work on interpersonal criticism, none of which is ever mentioned. As such, it appears that the authors either aren’t aware of that work (i.e. beyond emotion regulation, topics like reactance, face-threat etc. come to mind), or that they ignore it in order to keep the intro slim.. However, I found this resulted in an intro that was a bit bleak, be it in terms of the psychological processes or the neural mechanisms.

2) Data analysis/Results: My largest criticism refers to the results/analysis. Frankly, I didn’t fully understand what you did and how. I am not an fNIRS person, but I have training in fMRI/GLM as well as EEG methods, so this it not necessarily my lack of skill. For example, where you write: “… no sig main effect of OC ratings for any target relationship on hemodynamic changes in the PFC found”, I don’t really understand what analysis underlies that statement? Did you run PC ratings as a parametric regressor?

Same later: “After the general linear model was conducted, Person pm-correlations were calculated”. I get what all those words mean, but I couldn’t say what data from which conditions were used and compared. I understand that you might think this all is self-explanatory, but unless I am currently particularly dumb, the issue might be with your writing, i.e. that you’re so deep in the weeds of this study, but others cannot “see” what/where you are pointing to. I suggest that you try to rewrite this part and add figures or tables that can “hold the reader by the hand”.

Given that I had a hard time following the results, I cannot really say what they are: My impression is that there barely are clear results, and if that were true, it would pose a bigger problem of the paper. As said, I am sympathetic to the approach, very much so: However, given that the study clearly doesn’t rely on strong theory, it should i) show convincing null-finding or ii) report an interesting finding (that is marked as exploratory). Also, some sort of manipulation check of the neural data (e.g. a comparison criticism-message vs. control message) and generally, a better graphical illustration of the signal you measured (to demonstrate the reader how e.g. the event-related response to reading the vignette looked like) would be worthwhile. Also, I found the figures difficult to read and they could be optimized for clarity and information content.

Although I raise a number of criticism, I found the topic very interesting and the lab-part of the study seemed rigorously executed. As such, I hope that the authors will be able to address my concerns regarding the analysis/results. Also, I really liked that the data will be made available - that adds to the positive impression of this new approach.

6. PLOS authors have the option to publish the peer review history of their article (what does this mean?). If published, this will include your full peer review and any attached files.

Reviewer #1: No

Reviewer #2: No

---

## [Author Response · Author response to Decision Letter 0]

3 Jun 2020

Dear Reviewers, 

Thank you for your comments. Please find our response and revisions made to the manuscript based on your comments as highlighted in blue below. 

Reviewer #1: This manuscript describes an interesting study towards the neural correlates of different sources of observed criticism. Participants (N=49, undergraduate students) were asked to indicate their personal perception of criticism (PC), which was then associated with neural responses to several vignettes on observed criticism, as measured by fNRIS. Findings show a moderating effect of PC on neural responses: those who rated PC higher showed more dlPFC activation when reading vignettes about romantic partners and parents, but not friends.

Although the topic of the study is certainly interesting and a possible strong addition to the existing literature, the authors seem to switch between self-experienced criticism and observed criticism throughout the manuscript. This makes it difficult to understand what it is exactly that the authors have investigated, and how their findings contribute to existing literature. A reframing of the topic would certainly improve the manuscript. Below I have outlined several additional concerns and questions that I would like to see addressed in a revision of this manuscript:

Abstract

1. The sentence “Perceived criticism ratings for these relationships from 49 participants were collected” if formulated a bit odd and could be rephrased.

This has been rephrased to “49 participants were recruited and asked to rate the perceived criticism for these relationships.”

Introduction

2. The authors do a good job explaining the neural correlates of emotion regulation and social cognitive processing, but information on the neural correlates of emotion reaction is scarce. Additionally, what exactly is the role of emotion reaction in the described process from experiencing criticism to emotional responding?

As proposed in Lemerise & Arsenio (2000), the experience of emotion will affect social information processing and decision making in social situations. Hence, emotions elicited when experiencing criticism will play a role in the consequent emotion response. We have updated the discussion on the neural correlates of emotion reaction and the relationship between emotion reaction and the behavioural response as follows: 

“First, with regard to (i) emotion reaction, negative emotional reactions to criticism are considered normative as mentioned above [4] such as feelings of hurt. Negative emotional reactions have been associated with activity in both prefrontal regions such as the rostral anterior cingulate cortex (rACC) [13-15] and subcortical-limbic regions such as the amygdala [16-17]. There has also been some initial evidence pointing towards amygdala hyperactivation and prefrontal hypoactivation in individuals high on perceived criticism (PC) who listened to maternal criticism [18]. This pattern of activation is indicative of the implication of a neurocircuit related to dysfunctional emotional regulation and depression vulnerability [19-21]. Sustained brain activity in affective networks was also observed during the presentation of negative stimuli and subsequent rest periods [18, 22-23]. In addition, the negative emotional reaction accompanying the experience of criticism is likely to influence processing of social and emotional information (i.e. the criticism) and decision making in social situations [24]. The intensity of the emotional reaction can influence attentional deployment and meaning attributed to the situation, thereby cueing different response types to the encountered situation.”

3. The introduction mainly contains literature on how the experience of criticism can shape later relationships and how criticism can affect neural processing. Given that the participants did not experience criticism themselves but merely observed criticism on someone else, literature should be added to describe results from earlier studies on observed or third-party criticism.

We have included a brief discussion of observed criticism and work done on self/other referential processing in the Introduction as follows: 

“While most of the studies discussed above have been conducted on self-referential criticism, few studies have been conducted on the neural response during the observation of criticism occurring in the social interactions between others – other referential criticism. These few functional magnetic resonance imaging (fMRI) studies also focused only on examining the (i) medial PFC due to its engagement in self-processing and mentalising about others’ states and (ii) amygdala due to its influence on attention to emotional-expression stimuli as regions of interest. One study found that individuals with generalised social phobia showed significantly increased blood oxygen level dependent (BOLD) responses in the dorsal medial PFC and the amygdala compared to the control group in response to self-referential criticism but not to other-referential criticism [38]. Similar fMRI findings in another study have also indicated activations in prefrontal regions such as the ventral and dorsal medial PFC, posterior cingulate cortex, inferior parietal lobule and temporal poles during self-referential processing compared to other-referential processing of criticism stimuli [39]. In addition, studies comparing self-referential and other-referential processing of valenced stimuli have also found evidence of differences in the activation patterns between self and other-referential processing. Activation in the ventral and dorsal anterior medial PFC for trait adjective judgments targeting the self while activation in the posterior dorsal medial PFC was observed when adopting a third person perspective in making trait adjective judgments about another person [40]. Different activation patterns were also observed reading valenced trait adjectives associated with either the self or other [41].”

In relating findings from the current study to past work (refer to Item 16), we have also mentioned these earlier studies as follows: 

“In addition, findings from this study contributes to a greater understanding of the neural mechanisms involved in observing criticism from a third-person perspective and other-referential processing of criticism. Firstly, the significant interaction effect between relationship type and PC ratings indicate that the identity of the “other” implicated in other-referential processing can influence the neural response involved. In this respect, studies investigating self/other referential processing tend to either use generic third person pronouns (such as He/She; see [65] for the Visual-Verbal Self-Other Referential Processing Task) or only one specific type of “other” such as a close friend in [40]. Secondly, more areas of the PFC were studied in the present study, which suggest the possibility of other processes and prefrontal areas implicated in other-referential processing that have not been previously studied. Finally, understanding the processes involved when an individual observes criticism being directed at others can provide a clearer picture to how individuals navigate and respond during social interactions with others that involve occurrences of criticism directed at others (e.g. watching a friend’s parent criticise him/her).”

Methods

4. Were all participants recruited in the same way? As the authors describe “undergraduates compensated with course credits and undergraduates recruited through word of mouth and advertisements”

We have clarified and replaced the sentence “Participants (n = 50, mean age = 21.67, females = 25) were undergraduates recruited through word of mouth and advertisements and compensated with course credits or remuneration.”

5. Were participants also screened on underlying psychiatric disorders, given the strong link between perception of criticism and psychopathology?

The participants were not screened for psychiatric disorders. We have noted this accordingly in the limitations as follows: “In addition, participants in this study’s sample were not screened for psychiatric disorders. Future studies can look to investigate whether there are differences between individuals diagnosed with psychiatric disorders and healthy individuals in terms of processing criticism observed in the social interactions of others.”

6. First the authors state that vignettes were shown in the same order to all participants, but then it is explained that participants are randomly assigned to one of three groups with different orders in the vignettes. Does this mean that the vignettes were not shown in the same order to all participants?

The vignettes were shown in the same order to all participants but the description of the relationship of the protagonist and the source of criticism in each vignette was randomised as initially described “Vignettes were shown in the same order to all participants. However, participants were randomly assigned to 3 groups where the description of the relationship of the protagonist to the source of criticism for each vignette was randomised as: (a) Romantic Partner-Friends-Parents, (b) Friends-Parents-Romantic Partner, and (c) Parents-Romantic Partner-Friends.” 

However, we note that the explanation may not have been adequately clear. Hence, to clarify this point, we have updated this part to include a more detailed explanation as follows: 

“Three different vignettes were used in the study where each vignette described a different scenario of criticism in a social situation. The three vignettes were shown in the same order to all participants. Before each vignette was shown, a brief description of the relationship between the protagonist and the source of criticism was shown to the participants. The order in which the relationship described was presented for each scenario of criticism was counterbalanced as: (a) Romantic Partner-Friends-Parents, (b) Friends-Parents-Romantic Partner, and (c) Parents-Romantic Partner-Friends. Participants were then randomly assigned to groups where they viewed the vignettes with the accompanying relationships’ descriptions in one of these orders.” 

7. In the section “Questionnaires” the authors wrote “romantic partner, (ii) friend, parents; mother and father”. I assume parents should be a separate, third category.

We have edited the text accordingly: “(i) romantic partner, (ii) friend and (iii) parents; mother and father”. 

8. Table 1: correlations appear to be significant, but statistical information (p-value) is missing.

We have updated Table 1 with the corresponding p-values. 

9. Table 1: Are the correlation coefficients for mother and father significantly different from each other? If this is indeed the case: should this be taken into account, given that in the vignettes parents are collapsed (mother & father)?

We found that the PC ratings for mothers and fathers were significantly correlated (r = 0.578. In addition, we found that there were no significant differences between PC ratings (Mother) and PC ratings (Father) in the sample. 

10. Did every participant only see 3 vignettes (1 about romantic partner, 1 about friend, 1 about parents)? If so, how reliable are the fNIRS results if they are only based on one trial in each condition?

The fNIRS recording involves multiple sampling during each trial with a sampling rate of 7.81Hz and processed data involves block averaging of samples acquired during each trial. Generally, there are also a significant number of fNIRS studies conducted based on one trial per condition rather than multiple trials. 

11. What exact instructions did the participants receive? Were participants instructed to identify themselves with the protagonist of the vignette?

We have included the instructions given to participants in the Experimental Procedure section as follows: “Participants were informed that they would be reading vignettes involving interactions between people and a description of the relationship between the persons involved will be given.”

12. Given that amount of perceived criticism was also a measure of interest, did vignettes differ in level of criticism?

Vignettes were standardised as much as possible by adopting a common structure and length throughout the vignette as well as the criticism segment to ensure that they did not differ in level of criticism. We also found that there were no significant differences between vignettes in terms of participants’ ratings of the impact of the criticism (F(2, 144) = .002, p = .998). 

13. Were PC ratings (resulting from the questionnaires) also correlated with level of justification and impact of the criticism for each vignette?

PC ratings were not significantly correlated with the level of justification and impact of the criticism for each vignette. We have also included these correlations in a new table under the subsection on Experimental Questionnaire.

14. How was the 10-point scale for justification and impact formulated? (1 = not justified, 10= very justified)? And how can these scores indicate the authenticity of the vignettes?

Participants were asked to rate justification and impact of the criticism as a manipulation check because the reaction and perception of criticism are influenced by attributions or explanations thought to underlie the criticism, where perceptions of intentionality and fairness would result in different reactions and intensity of reactions. Interpretations of another’s behaviour play a role in determining whether a behaviour is perceived as critical. Hence, justification would be indicative the extent to which participants thought that the criticism in the vignette could be explained and thereby, perceived to be criticism. Additionally, the impact of the criticism would be indicative of the extent to which participants perceived the vignette as criticism and the extent to which they felt that it would affect the protagonist. Thus, by asking participants to rate the vignettes on justification and impact, we can check the extent to which participants viewed scenarios described in the vignette to be criticism. 

Results

15. What exactly was investigated in the Pearson product-moment correlations? PC from Mother with what?

The Pearson product-moment correlations were calculated between PC (Mother) and the beta coefficients for each relationship type calculated from the individual level GLM. We have clarified this in the paragraph on the Fisher r-to-z transformation as follows: “In examining the interaction between relationship type and PC rating of mothers, the Pearson product-moment correlations were calculated between beta coefficients for each relationship type and PC rating of mothers (Table 3). The correlation coefficients are as follows: (i) Romantic Partner-PC (Mother) (RPM) was (r = 0.20, n = 44), (ii) Friend-PC (Mother) (FPM) was (r = -0.44, n = 43), and (iii) Parents-PC (Mother) (PPM) was (r = 0.26, n = 41). To examine the significance of the difference between the correlation coefficients, Fisher r-to-z transformation was applied in order to compare the z scores (Table 3). The results summarised in Table 3 showed significantly different correlations for (i) the Romantic Partner-PC (Mother) and Friend-PC (Mother) correlations (Z = 3.02, p < 0.05) and (ii) Friend-PC (Mother) and Parents-PC (Z = -3.25, p < 0.05). The Romantic Partner-PC and Parents-PC correlations were not significantly different (Z = -0.29, p = 0.77 > 0.05). From Fig 5, it can be observed that as PC ratings for the individual’s mother increased, activation of the left middle frontal gyrus of the dlPFC (BA46L) increased when reading the vignettes describing criticism from romantic partners and parents but decreased when reading the vignettes describing criticism from friends.“

Discussion

16. Throughout the manuscript the authors seem to switch between self-experienced criticism and observed criticism. It should be made clearer what processes were actually studied and how the findings from the current study can be related to earlier work on self-experienced criticism.

We have gone through the manuscript and clarified in each instance which (self-directed or observed criticism) we are referring to when discussing results from the current study and previous work. 

We have also discussed in greater detail in the Discussion how findings from the current study can be related to earlier work as follows: 

“In addition, findings from this study contributes to a greater understanding of the neural mechanisms involved in observing criticism from a third-person perspective and other-referential processing of criticism. Firstly, the significant interaction effect between relationship type and PC ratings indicate that the identity of the “other” implicated in other-referential processing can influence the neural response involved. In this respect, studies investigating self/other referential processing tend to either use generic third person pronouns (such as He/She; see [65] for the Visual-Verbal Self-Other Referential Processing Task) or only one specific type of “other” such as a close friend in [40]. Secondly, more areas of the PFC were studied in the present study, which suggest the possibility of other processes and prefrontal areas implicated in other-referential processing that have not been previously studied. Finally, understanding the processes involved when an individual observes criticism being directed at others can provide a clearer picture to how individuals navigate and respond during social interactions with others that involve occurrences of criticism directed at others (e.g. watching a friend’s parent criticise him/her).”

17. A final concluding paragraph is missing.

This has been added in this version of the manuscript as follows: 

“In conclusion, findings from the present study suggest that criticism perceived in an individual’s interpersonal environment can influence how one processes criticism observed from a third-party perspective. This provides preliminary evidence that points toward differences in social information processing as a function of one’s own interpersonal environment and past experiences, specifically in the context of observing criticism as it occurs in social interactions of others. Given the multitude of factors that shape an individual’s interpersonal environment and the experiences one derives from this environment, it follows that there remains much to be elucidated about how such factors and individual differences would influence social information processing and consequently, how one views and processes observations of criticism as they occur in the interactions of others.”

 

Reviewer #2: Reviewer Form - The reviewer form is broken into the following parts:

In their article Disapproval from romantic partners, friends and parents: source of criticism regulates prefrontal cortex activity the authors report results from a study in which they used fNIRS to measure the brain response to criticizing messages (vignettes) coming from different (hypothetical) sources.

The topic of the paper is interesting and I believe this type of research is timely, the approach innovative, and the research has generally been conducted well. However, I had some difficulty to understand the design and results, and that made it really difficult for me to evaluate the manuscript. The authors might be able to address this in a revision, although I am not completely sure (since I didn’t fully understand the results).

1) Theoretical embedding/strategy: As said, I am generally favorable towards the approach. The use of fNIRS is novel and has potential. Also, I am aware that working in a novel area often goes along with a lack of prior theoretical work, so necessarily you have to go a bit of a data-driven route. That is fine with me, although I do think it would be worthwhile to dig a little deeper into the work that exists on these topics. E.g. in the communication science literature there has been a lot of work on interpersonal criticism, none of which is ever mentioned. As such, it appears that the authors either aren’t aware of that work (i.e. beyond emotion regulation, topics like reactance, face-threat etc. come to mind), or that they ignore it in order to keep the intro slim.. However, I found this resulted in an intro that was a bit bleak, be it in terms of the psychological processes or the neural mechanisms.

We have included relevant discussions of the literature on interpersonal criticism and face threat detailed below: 

“As such, the attributions made regarding the nature of the criticism that can influence an individual’s reaction to the criticism. How another person’s behaviour is appraised is central to both an individual’s emotions, such as feelings of hurt in the case of criticism, and the perceived impact of the behaviour on the relationship. For example, individuals who perceive a particular comment to be intentionally hurtful distanced themselves from the source of criticism and hurt as well as felt greater emotional pain than if the comment were to perceived to be unintentional [59]. In a similar vein, it has been argued that the nature of criticism as a face threatening act will also influence one’s reaction. According to the face management theory proposed by [60], face is the desired social image created by an individual through interactions with others. A relationship-specific face [61] is enacted where an individual’s face becomes inextricably bound in a shared relationship identity as relationships develop closeness. As a result, expressions of disapproval pose a threat to one’s relationship specific face, potentially causing hurt feelings and perceived relational devaluation. A study found that perceptions of face threat were associated with emotional reactions such as feelings of anger, hurt, embarrassment and anxiety/depression as well as perceptions of fairness [62]. Additionally, findings from this study also suggested that the nature and rules governing different social relationships affected the relevance of negative face threats – threat to one’s need for autonomy and desire to avoid impositions by others - for friends than for romantic partners. The study found that perceived threat to negative threat were associated with feelings of anger/hurt and damage to the relationship only in friends but not romantic partners [62]. Hence taking together differences in attributions, levels of perceived face threat and qualities of interpersonal relationships, we expect that individuals differing in PC may also have different responses to criticism originating from different sources.”

2) Data analysis/Results: My largest criticism refers to the results/analysis. Frankly, I didn’t fully understand what you did and how. I am not an fNIRS person, but I have training in fMRI/GLM as well as EEG methods, so this it not necessarily my lack of skill. For example, where you write: “… no sig main effect of OC ratings for any target relationship on hemodynamic changes in the PFC found”, I don’t really understand what analysis underlies that statement? Did you run PC ratings as a parametric regressor?

PC ratings were the between subject factor in a two-way, mixed ANOVA that was run on the processed HbO data. 

Same later: “After the general linear model was conducted, Person pm-correlations were calculated”. I get what all those words mean, but I couldn’t say what data from which conditions were used and compared. I understand that you might think this all is self-explanatory, but unless I am currently particularly dumb, the issue might be with your writing, i.e. that you’re so deep in the weeds of this study, but others cannot “see” what/where you are pointing to. I suggest that you try to rewrite this part and add figures or tables that can “hold the reader by the hand”.

We have rewritten this paragraph on the Fisher r-to-z transformation and referenced the relevant tables and figures accordingly as follows: “In examining the interaction between relationship type and PC rating of mothers, the Pearson product-moment correlations were calculated between beta coefficients for each relationship type and PC rating of mothers (Table 4). The correlation coefficients are as follows: (i) Romantic Partner-PC (Mother) (RPM) was (r = 0.20, n = 44), (ii) Friend-PC (Mother) (FPM) was (r = -0.44, n = 43), and (iii) Parents-PC (Mother) (PPM) was (r = 0.26, n = 41). Fisher r-to-z transformation was applied to the correlation coefficients in order to examine the significance of the difference between the correlation coefficients by comparing the z scores (Table 4). The results summarised in Table 4 showed significantly different correlations for (i) the Romantic Partner-PC (Mother) and Friend-PC (Mother) correlations (Z = 3.02, p < 0.05) and (ii) Friend-PC (Mother) and Parents-PC (Z = -3.25, p < 0.05). The Romantic Partner-PC and Parents-PC correlations were not significantly different (Z = -0.29, p = 0.77 > 0.05). From Fig 5, it can be observed that as PC ratings for the individual’s mother increased, activation of the left middle frontal gyrus of the dlPFC (BA46L) increased when reading the vignettes describing criticism from romantic partners and parents but decreased when reading the vignettes describing criticism from friends.“

Given that I had a hard time following the results, I cannot really say what they are: My impression is that there barely are clear results, and if that were true, it would pose a bigger problem of the paper. As said, I am sympathetic to the approach, very much so: However, given that the study clearly doesn’t rely on strong theory, it should i) show convincing null-finding or ii) report an interesting finding (that is marked as exploratory). Also, some sort of manipulation check of the neural data (e.g. a comparison criticism-message vs. control message) and generally, a better graphical illustration of the signal you measured (to demonstrate the reader how e.g. the event-related response to reading the vignette looked like) would be worthwhile. Also, I found the figures difficult to read and they could be optimized for clarity and information content.

The fNIRS data processing for each condition (i.e. recording during the reading of each vignette) takes a baseline measurement by including data recorded 5 seconds before the onset of the stimulus. While a control message was not included, fNIRS studies generally look to compare patterns of activation between conditions rather than with a control condition. 

We have included an example of the processed fNIRS signal in a new figure in the subsection on NIRS data pre-processing (Figure 4). We have also updated Figure 2 for improved clarity. 

Although I raise a number of criticism, I found the topic very interesting and the lab-part of the study seemed rigorously executed. As such, I hope that the authors will be able to address my concerns regarding the analysis/results. Also, I really liked that the data will be made available - that adds to the positive impression of this new approach.

---

## [Decision Letter · Decision Letter 1]

13 Aug 2020

PONE-D-20-03038R1

Disapproval from romantic partners, friends and parents: source of criticism regulates prefrontal cortex activity

PLOS ONE

Dear Dr. Esposito,

Thank you for submitting your manuscript to PLOS ONE. After careful consideration, we feel that it has merit but does not fully meet PLOS ONE’s publication criteria as it currently stands. Therefore, we invite you to submit a revised version of the manuscript that addresses the points raised during the review process.

We look forward to receiving your revised manuscript.

Kind regards,

Wi Hoon Jung, PhD

Academic Editor

PLOS ONE

Reviewers' comments:

Reviewer's Responses to Questions

**Comments to the Author**

1. If the authors have adequately addressed your comments raised in a previous round of review and you feel that this manuscript is now acceptable for publication, you may indicate that here to bypass the “Comments to the Author” section, enter your conflict of interest statement in the “Confidential to Editor” section, and submit your "Accept" recommendation.

Reviewer #1: (No Response)

Reviewer #3: (No Response)

2. Is the manuscript technically sound, and do the data support the conclusions?

Reviewer #1: Partly

Reviewer #3: Partly

3. Has the statistical analysis been performed appropriately and rigorously? 

Reviewer #1: Yes

Reviewer #3: Yes

4. Have the authors made all data underlying the findings in their manuscript fully available?

Reviewer #1: Yes

Reviewer #3: Yes

5. Is the manuscript presented in an intelligible fashion and written in standard English?

Reviewer #1: Yes

Reviewer #3: Yes

6. Review Comments to the Author

Reviewer #1: Most of my comments have been sufficiently addressed. Below I formulated some remaining issues, most in relation to the processes that the authors describe throughout the manuscript (i.e. self-experienced criticism) compared to what the authors actually investigate (i.e. observed criticism):

Abstract

1. Only in the last three lines of the abstract it becomes clear that the authors actually investigated observed/third-party criticism, instead of self-experienced criticism. This should be clarified from the beginning.

Introduction

1. Although the authors have done a good job in expanding the introduction to include

literature on neural correlates of emotion reaction and observed criticism, the introduction still largely describes processes involved in personal experiences of criticism, which is not what the authors actually investigated. For example, the expectations of high vs low PC individuals seem to be formulated for self-experienced criticism, not observed criticism. It should be made clearer what process the authors actually investigated.

1. I like how the authors have linked the findings from their study to previous work in the discussion. However, this would be good to also mention in the introduction, as to explain why it is interested to study (neural correlates of) observed criticism in the first place.

Results

2. Following comments in the earlier review, the authors have done a good job in adapting the results section to “In examining the interaction between relationship type and PC rating of mothers, the Pearson product-moment correlations were calculated between beta coefficients for each relationship type and PC rating of mothers (Table 3). The correlation coefficients are as follows: (i) Romantic Partner-PC (Mother) (RPM) was (r = 0.20, n = 44), (ii) Friend-PC (Mother) (FPM) was (r = -0.44, n = 43), and (iii) Parents-PC (Mother) (PPM) was (r = 0.26, n = 41). To examine the significance of the difference between the correlation coefficients, Fisher r-to-z transformation was applied in order to compare the z scores (Table 3). The results summarised in Table 3 showed significantly different correlations for (i) the Romantic Partner-PC (Mother) and Friend-PC (Mother) correlations (Z = 3.02, p < 0.05) and (ii) Friend-PC (Mother) and Parents-PC (Z = -3.25, p < 0.05). The Romantic Partner-PC and Parents-PC correlations were not significantly different (Z = -0.29, p = 0.77 > 0.05). From Fig 5, it can be observed that as PC ratings for the individual’s mother increased, activation of the left middle frontal gyrus of the dlPFC (BA46L) increased when reading the vignettes describing criticism from romantic partners and parents but decreased when reading the vignettes describing criticism from friends.”

Although this adaptation makes this specific section easier to understand it might still be helpful if short interpretations would be provided. For example, the results in Table 3 show significant different correlations for several relationships; how should these findings be interpreted, what do they actually mean for your study?

Reviewer #3: The present manuscript investigates using functional Near Infrared Spectroscopy (fNIRS) prefrontal (PFC) responses of 49 participants to perceptual criticism from romantic partners, friends and parents (mother and father). The authors reveal an enhancement of the left dorsolateral PFC (DLPFC) links to criticism from romantic partners. At the opposite, a decrease of the left DLPFC is observed for criticism from friends.

The authors already assessed several concerns of reviewer 1. However, I have still important recommendations regarding the methodology.

Major concerns

1. The authors described in the introduction the importance of psychiatric disorders on perceptual criticism. Yet, they did not exclude participants with such disorders in the present study. Even if the authors considered this point in the discussion, as a limitation, I think it is a real issue here. If few participants had psychiatric or neurological disorders, the results observed could be totally biased. I strongly recommend to the authors to contact a posteriori the 49 participants to ensure that all are healthy. If not, the authors should exclude the "non-healthy" participants from the analysis.

2. The experimental paradigm only included one trial per condition. Even if each trial was presented during 90 seconds, the data acquisition was quiet low for a fNIRS study (+/- 7 Hz). I am not sure if statistically speaking, the number of data are sufficient to generalized the results. The authors argued that is a common experimental paradigm in fNIRS, however, such protocols are now scarce in the field... Yet, I have to admit that the statistics performed later by the authors seem robust and relevant.

3. The authors indicated "multi-distant channel setup" in the section fNIRS recording. To my understanding, the inter-distance probes varied depending on the channels. If yes, the authors have to specify the range of these variations (not just 3cm max as mentioned in the manuscript) and they also need to justify their choice. If the distances are not correct, the fNIRS data cannot take it seriously...

Minor concerns

4. Figure 1 do not correspond to the experimental setup. The authors investigated PFC activation using 20 channels, yet the figure represent a whole-brain fNIRS device with 100 channels.

5. Table 1 indicates the correlations between perceptual criticism rating and target relationship. Yet, the authors did not mention the type of correlation at this point of the manuscript.

6. Figure 3 indicates channel locations. How the authors determined the exact location of the channels? EEG standard, 3D digitizer? The authors need to add the information below the figure.

7. L.358 "replacement of spike artefacts with nearest or random signals". What are the rules to determine such choices?

8. L.438 - 444. The authors cannot interpret the data without any statistics. However, If my understanding is correct, the authors add a similar paragraph with the corresponding stats and figure later L.452 - 459. The authors should thus remove the first paragraph.

If the authors can assess all my concerns, the present manuscript should improve our knowledge on the field. Use fNIRS as a methodological approach to investigate perceptual criticism - PFC activation is also interesting.

7. PLOS authors have the option to publish the peer review history of their article (what does this mean?). If published, this will include your full peer review and any attached files.

Reviewer #1: No

Reviewer #3: No

---

## [Author Response · Author response to Decision Letter 1]

15 Aug 2020

Dear Reviewers, 

Thank you for your comments. Please find our response and revisions made to the manuscript based on your comments as highlighted in blue below. 

Reviewer #1: Most of my comments have been sufficiently addressed. Below I formulated some remaining issues, most in relation to the processes that the authors describe throughout the manuscript (i.e. self-experienced criticism) compared to what the authors actually investigate (i.e. observed criticism):

Abstract

1. Only in the last three lines of the abstract it becomes clear that the authors actually investigated observed/third-party criticism, instead of self-experienced criticism. This should be clarified from the beginning.

Thank you for pointing this out. We have included a clarification in the beginning of the abstract as follows: “The present study aims to investigate neural responses to observing criticism occurring in the context of originating from three different relationship types: romantic partners, friends, and parents – from a third-party perspective.”

Introduction

1. Although the authors have done a good job in expanding the introduction to include

literature on neural correlates of emotion reaction and observed criticism, the introduction still largely describes processes involved in personal experiences of criticism, which is not what the authors actually investigated. For example, the expectations of high vs low PC individuals seem to be formulated for self-experienced criticism, not observed criticism. It should be made clearer what process the authors actually investigated.

Thank you for your comment on the revisions to the Introduction. We have further clarified on this by specifying in the Introduction as follows: 

“Given involvement of these parts of the PFC in emotion reactivity and regulation, coupled with previous findings of decreased prefrontal control to criticism, the present study sought to investigate the neural correlates of responses to other referential criticism specifically in the PFC using functional near-infrared spectroscopy (fNIRS).”

“Hence taking together differences in attributions, levels of perceived face threat and qualities of interpersonal relationships, we expect that individuals differing in PC may also have different responses to other referential criticism originating from different sources.”

“As mentioned, majority of neuroimaging studies discussed above were investigating self-referential criticism with few studies looking at the neural correlates of other referential criticism. In addition, the studies on other referential criticism tended to look primarily at the mPFC as a region of interest and varied in terms of the identity of the “other” as the source of the other referential criticism. In the present study, we aim to investigate PFC activation during exposure to other referential criticism involving an “other” from three different relationship types – romantic partners, friends and parents.

Hence, the present study aims to contribute to existing literature by providing both novel fNIRS data of PFC activation during exposure to other referential criticism and a functional characterization of the mental processes underlying the neural response to criticism observed in the social interactions of others from a third-party perspective while comparing romantic partners, friends, and parents as sources of criticism.”

1. I like how the authors have linked the findings from their study to previous work in the discussion. However, this would be good to also mention in the introduction, as to explain why it is interested to study (neural correlates of) observed criticism in the first place.

Thank you for your comment on the revisions to the Discussion. We have included mention of this in the Introduction as follows: “As mentioned, majority of neuroimaging studies discussed above were investigating self-referential criticism with few studies looking at the neural correlates of other referential criticism. In addition, the studies on other referential criticism tended to look primarily at the mPFC as a region of interest and varied in terms of the identity of the “other” as the source of the other referential criticism. In the present study, we aim to investigate PFC activation during exposure to other referential criticism involving an “other” from three different relationship types – romantic partners, friends and parents.

Hence, the present study aims to contribute to existing literature by providing both novel fNIRS data of PFC activation during exposure to other referential criticism and a functional characterization of the mental processes underlying the neural response to criticism observed in the social interactions of others from a third-party perspective while comparing romantic partners, friends, and parents as sources of criticism.”

Results

2. Following comments in the earlier review, the authors have done a good job in adapting the results section to “In examining the interaction between relationship type and PC rating of mothers, the Pearson product-moment correlations were calculated between beta coefficients for each relationship type and PC rating of mothers (Table 3). The correlation coefficients are as follows: (i) Romantic Partner-PC (Mother) (RPM) was (r = 0.20, n = 44), (ii) Friend-PC (Mother) (FPM) was (r = -0.44, n = 43), and (iii) Parents-PC (Mother) (PPM) was (r = 0.26, n = 41). To examine the significance of the difference between the correlation coefficients, Fisher r-to-z transformation was applied in order to compare the z scores (Table 3). The results summarised in Table 3 showed significantly different correlations for (i) the Romantic Partner-PC (Mother) and Friend-PC (Mother) correlations (Z = 3.02, p < 0.05) and (ii) Friend-PC (Mother) and Parents-PC (Z = -3.25, p < 0.05). The Romantic Partner-PC and Parents-PC correlations were not significantly different (Z = -0.29, p = 0.77 > 0.05). From Fig 5, it can be observed that as PC ratings for the individual’s mother increased, activation of the left middle frontal gyrus of the dlPFC (BA46L) increased when reading the vignettes describing criticism from romantic partners and parents but decreased when reading the vignettes describing criticism from friends.”

Although this adaptation makes this specific section easier to understand it might still be helpful if short interpretations would be provided. For example, the results in Table 3 show significant different correlations for several relationships; how should these findings be interpreted, what do they actually mean for your study?

We have included a short interpretation as follows: “This significant result suggests that PC ratings moderate neural response to criticism originating from different sources of different relationship types, resulting in different activation patterns observed between high and low PC individuals for criticism from the different relationship types.” 

Reviewer #3: The present manuscript investigates using functional Near Infrared Spectroscopy (fNIRS) prefrontal (PFC) responses of 49 participants to perceptual criticism from romantic partners, friends and parents (mother and father). The authors reveal an enhancement of the left dorsolateral PFC (DLPFC) links to criticism from romantic partners. At the opposite, a decrease of the left DLPFC is observed for criticism from friends.

The authors already assessed several concerns of reviewer 1. However, I have still important recommendations regarding the methodology.

Major concerns

1. The authors described in the introduction the importance of psychiatric disorders on perceptual criticism. Yet, they did not exclude participants with such disorders in the present study. Even if the authors considered this point in the discussion, as a limitation, I think it is a real issue here. If few participants had psychiatric or neurological disorders, the results observed could be totally biased. I strongly recommend to the authors to contact a posteriori the 49 participants to ensure that all are healthy. If not, the authors should exclude the "non-healthy" participants from the analysis.

We acknowledge that we were not clear in our previous revision – while the participants were not screened as part of our experimental procedure, the participants were recruited from a research participant pool. The research participant pool itself has an exclusion criteria – participants with psychiatric disorders are excluded. 

We have clarified this accordingly in the Recruitment subsection as follows: “Having a psychiatric disorder was an exclusion criteria for participants.“ Similarly, we have also clarified in the Limitations regarding future studies as follows: “Although the present study was conducted only on healthy individuals, future studies can look to investigate whether there are differences between individuals diagnosed with psychiatric disorders, such as mood disorders and social anxiety, and healthy individuals in terms of processing criticism observed in the social interactions of others.” 

2. The experimental paradigm only included one trial per condition. Even if each trial was presented during 90 seconds, the data acquisition was quiet low for a fNIRS study (+/- 7 Hz). I am not sure if statistically speaking, the number of data are sufficient to generalized the results. The authors argued that is a common experimental paradigm in fNIRS, however, such protocols are now scarce in the field... Yet, I have to admit that the statistics performed later by the authors seem robust and relevant.

Thank you for your consideration of our statistical analysis as robust and relevant. We are aware that fNIRS devices have lower sampling resolution compared to other devices such as the EEG. However, we would like to highlight that the signal measured by the fNIRS (that is, the BOLD signal extracted from levels of oxyHb) has a cycle of 7-10 seconds. In addition, majority of commercially available fNIRS devices in the market have a similar sampling rate (see Quaresima & Ferrari, 2019 which notes that “transportable/wireless commercially available fNIRS systems have a time resolution of 1 to 10 Hz”) and even if the sampling rate ~7 Hz is “low”, it is still detailed enough to capture the variable that we intend to measure. Furthermore, we agree that fNIRS studies in this specific field are still scarce. However, the number of studies is increasing and we are following a standard procedure with the NIRSport device we used (for example, see Vrana et al, 2016 which used the same NIRSport device with the same signal sampling rate of 7.81Hz as well). Finally, all the data have also been made available online in our data repository – this allows other groups to use our data for further analysis or replication purposes. 

3. The authors indicated "multi-distant channel setup" in the section fNIRS recording. To my understanding, the inter-distance probes varied depending on the channels. If yes, the authors have to specify the range of these variations (not just 3cm max as mentioned in the manuscript) and they also need to justify their choice. If the distances are not correct, the fNIRS data cannot take it seriously...

There has been a long debate in the field of fNIRS and EEG on this point. We are using a standard NIRS cap that fits the scalp of the participants. The software that we are using (as with other software available) for the fNIRS data acquisition loads a montage of the channel setup (placement of each emitter/detector) and overcomes this issue by automatically correcting for the interoptode distance based on this pre-loaded montage.

Minor concerns

4. Figure 1 do not correspond to the experimental setup. The authors investigated PFC activation using 20 channels, yet the figure represent a whole-brain fNIRS device with 100 channels.

Thank you for this comment and we have elaborated further in the Figure 1 caption to provide a clearer description of Figure 1 as follows: “Digital rendering of experimental setup depicting (i) 20 channel setup, (ii) NIRS device and (iii) laptop placement.” Figure 1 is a digital rendering of a photo of the actual experimental setup. The cap seen in the figure is a standard NIRS cap where the montage we used was the one described involving 8 emitters and 7 detectors forming the 20 channel setup. 

5. Table 1 indicates the correlations between perceptual criticism rating and target relationship. Yet, the authors did not mention the type of correlation at this point of the manuscript.

Thank you for pointing this out - we have included this sentence in the paragraph preceding Table 1 to make it clearer: ” Table 1 indicates the correlations between the PC ratings of the three relationship types.”

6. Figure 3 indicates channel locations. How the authors determined the exact location of the channels? EEG standard, 3D digitizer? The authors need to add the information below the figure.

A standard cap was used where placement of the emitters/detectors on the cap was based on a paired montage in the software which is based on the EEG 10-20 system. We have included this information in the Figure 3 caption as follows: “Montage for probe placement is based on the 10-20 system.”

7. L.358 "replacement of spike artefacts with nearest or random signals". What are the rules to determine such choices?

The software replaces the signal with the nearest signals where possible; otherwise, if there are more adjacent spikes for which replacement with nearest signals is not obvious to determine, it replaces the spike with random signals. However, we would like to stress that participants with significantly “noisy” data were excluded (as noted in the Results section that “Data for one additional male participant were excluded at the pre-processing stage due to poor signal quality.”). Hence, the data of the remaining participants included in the analysis had < 0.01% of spike artefacts. 

8. L.438 - 444. The authors cannot interpret the data without any statistics. However, If my understanding is correct, the authors add a similar paragraph with the corresponding stats and figure later L.452 - 459. The authors should thus remove the first paragraph.

We have removed the first paragraph (L438-444). 

If the authors can assess all my concerns, the present manuscript should improve our knowledge on the field. Use fNIRS as a methodological approach to investigate perceptual criticism - PFC activation is also interesting.

We hope we have addressed all your concerns. Thank you for your comments that have helped in making our article significantly clearer.

---

## [Decision Letter · Decision Letter 2]

21 Sep 2020

Disapproval from romantic partners, friends and parents: source of criticism regulates prefrontal cortex activity

PONE-D-20-03038R2

Dear Dr. Esposito,

We’re pleased to inform you that your manuscript has been judged scientifically suitable for publication and will be formally accepted for publication once it meets all outstanding technical requirements.

Kind regards,

Wi Hoon Jung, PhD

Academic Editor

PLOS ONE

Reviewers' comments:

Reviewer's Responses to Questions

**Comments to the Author**

1. If the authors have adequately addressed your comments raised in a previous round of review and you feel that this manuscript is now acceptable for publication, you may indicate that here to bypass the “Comments to the Author” section, enter your conflict of interest statement in the “Confidential to Editor” section, and submit your "Accept" recommendation.

Reviewer #3: All comments have been addressed

2. Is the manuscript technically sound, and do the data support the conclusions?

Reviewer #3: Yes

3. Has the statistical analysis been performed appropriately and rigorously? 

Reviewer #3: Yes

4. Have the authors made all data underlying the findings in their manuscript fully available?

Reviewer #3: Yes

5. Is the manuscript presented in an intelligible fashion and written in standard English?

Reviewer #3: Yes

6. Review Comments to the Author

Reviewer #3: The authors fully addressed all my comments and did a good job. I recommend the acceptation of this paper in PLOS ONE

7. PLOS authors have the option to publish the peer review history of their article (what does this mean?). If published, this will include your full peer review and any attached files.

Reviewer #3: **Yes: **Coralie Debracque

---

## [Editor Report · Acceptance letter]

24 Sep 2020

PONE-D-20-03038R2 

Disapproval from romantic partners, friends and parents: source of criticism regulates prefrontal cortex activity 

Dear Dr. Esposito:

I'm pleased to inform you that your manuscript has been deemed suitable for publication in PLOS ONE. Congratulations! Your manuscript is now with our production department. 

Kind regards, 

on behalf of

Dr. Wi Hoon Jung 

Academic Editor

PLOS ONE